# The 2030 Challenge in the Quality of Higher Education: Metacognitive, Motivational and Structural Factors, Predictive of Written Argumentation, for the Dissemination of Sustainable Knowledge

**Rosario Arroyo González**, **Javier de la Hoz-Ruiz** * and **Jesús Montejo Gámez**

Department of Didactics and School Organization, University of Granada, 18011 Granada, Spain;
rarroyo@ugr.es (R.A.G.); jmontejo@ugr.es (J.M.G.)

* Correspondence: delahoz96@correo.ugr.es; Tel.: +34-626716181

**Abstract:** The United Nations 2030 agenda includes quality university education, highlighting the importance of writing competence, as a basic skill for the dissemination of sustainable knowledge. However, there is little evidence of the factors that predict effective written communication to support such quality. Among these factors, the literature highlights motivation and writing metacognition, as well as the adequate structuring of the academic and/or scientific genre. The main novelty of the present research is the study of the relationships between the mentioned factors, measured with validated instruments. To this end, content analysis is first applied to determine the rhetorical moves of argumentative essays written by a sample of 72 university students. Secondly, the correlations between each of the rhetorical moves, metacognition and argumentative writing self-efficacy are calculated. The relationships are studied in depth, applying step-by-step linear regression models. Finally, the dependence of the results, observed with respect to unmeasured factors, is contrasted by means of a confirmatory analysis based on structural equations. The analyses show that it is the practical ability to express rhetorical moves—Conclusion and Bibliographic References—which predicts a students' writing metacognition. Moreover, the minor relationship that argumentative self-efficacy shows with the expression of rhetorical moves, compared to writing metacognition, point to the need to consider another motivational dimension that is driving the learning of the argumentative essay at university level, a hypothesis that is confirmed with the structural equations model. These, and other findings, allow for the establishment of a series of educational quality criteria for the empowerment of written argumentation in academic and scientific contexts.

**Keywords:** high-quality education; 21st century skills; argumentative essay; prediction factors; writing metacognition; rhetorical moves; writing self-efficacy

## 1. Introduction

In the Resolution approved by the General Assembly on September 25, 2015 [1], the United Nations approved the 2030 agenda for sustainable development, where 17 objectives and 169 targets were established. Target 4 aims to ensure inclusive and equitable quality education, as well as promoting lifelong learning opportunities for all. More specifically, target 4.3 aims to ensure, by 2030, equal access for all men and women to technical, professional and quality higher education, including university education. Thus, target 4.3. aims to create the didactical conditions that guarantee high-quality university education, as well as equal access for all to this higher education, which is

mainly professional and scientific. The quality of higher education includes attention to diversity, not only of gender but also of culture. Nevertheless, what other conditions must university education fulfil in order to achieve the quality requested by the United Nations?

In order to achieve this educational challenge, it is important to establish what is meant by quality of education and how to create the conditions to make it possible for everyone.

The quality of education, in the first place, establishes as a goal of its action the improvement of cognitive abilities, motivations and communication of all citizens, to enable a culture of peace and cooperation, within sustainable global development [2–4]. Therefore, the quality of education proposes the design of didactic models that provide logistical and material support; that is, sufficient instructions and resources adapted to diversity, in order to guide the learning of the previously described essential skills.

The definition of quality of education presented above highlights the importance of the equal access to communication skills for peace and cooperation. Thus, quality in higher education is necessarily linked to the cooperative development of operations, skills and/or communication strategies that produce concrete action for peace in the future and which is evaluated with criteria of efficiency and relevance, with equal opportunities for diversity [5]. Therefore, guaranteeing quality in university education means guaranteeing the learning of communicative competences, which is always influenced by cognitive, motivational and social factors. Additionally, when dealing with verbal communicative competences, linguistic factors cannot be forgotten.

This paper is then focused in guaranteeing the quality of higher education by promoting learning of communication skills. These define scientific and professional activities, whatever their characteristics of diversity. That is, verbal communication competencies and, more specifically, written verbal communication competencies. The latter refer to efficient and appropriate symbolic behaviours of interpersonal relationships (following grammar and culture rules) in linguistically diverse or multilingual contexts [6]. Therefore, written communication is always oriented towards an audience with whom one shares objectives, which can be formulated on an equal basis to build peace in a cooperative manner and ensure sustainable development.

Based on this vision of quality in education, written communication, in connection with other skills, would be an inalienable part of any quality educational model and, especially, in higher education [7]. This is so because writing is a basic skill in all institutional, governmental and administrative activities that govern employment and entrepreneurship. More specifically, written argumentation is configured as a decisive skill for the dissemination of sustainable knowledge in the academic, scientific, technical and business fields. Here, "sustainable knowledge" refers that one which has clear implications in the development of models of social and ecological sustainability and, thus, contributes significantly to problem solving [8].

In this context, it is also highlighted that argumentative written communication at university is a process in which metacognitive, motivational and structural factors interact to achieve scientific dissemination objectives within a disciplinary field [9,10], applying argumentative literary genre [11]. Therefore, to support the learning of written argumentative competence at university, which guarantees the quality of higher education, it is necessary to investigate the relationships between metacognitive, motivational and textual structuring factors, as well as the didactic implications of these relationships. To this aim, each of these factors and the related educational needs that have been identified in the international university context must first be defined.

It is understood that metacognition is the knowledge of general strategies for learning and thinking, as well as when and why these strategies are used. In addition, metacognition implies knowledge about oneself, in relation to the cognitive and motivational components. Motivation is understood to be those aspects that encourage the carrying out of certain actions and maintaining the conduct firm until all the objectives set by the subject are achieved. Finally, the structural factors focus on how a community of knowledge organises its written discourse, using certain strategies for shared purposes [12–14]. At the same time that the importance of these factors in effective

written communication is highlighted, deficiencies have been identified, such as lack of knowledge of self-regulation and sociocultural processes in writing [15], absence of correlation between motivational factors and writing performance [16] and inadequate structuring of the argumentative essays [17].

Therefore, this research aims to demonstrate the relationships that exist between different factors that determine the effectiveness of written argumentative communication, that is: (a) the way written argumentation is organised, (b) the metacognition of the writing process itself; and (c) motivation when facing the task of composing an argumentative text. All this is in order to offer a conceptual model that enables guidance, with quality criteria, of the argumentative learning of the writer at these levels. In order to do so we have set out to answer the following questions: Is there a relationship between the way of structuring the argumentative essay, writing metacognition and writing argumentative self-efficacy, by university students? Is it possible to identify any of these variables as factors that explain the others and/or anticipate new factors? What are the implications for a quality education at university that are inferred from the relationships found?

Beforehand, the background is laid out, in relation to the three variables mentioned.

## 1.1. Structuring the Scientific Argumentative Essay

In the current context of academic communication, the theory of textual genre [12] focuses on how a community of knowledge structures its written discourse using certain strategies for shared purposes. This theory is applied in the analysis of rhetorical moves to identify their function in written discourses and the subsequent classification of moves in steps. This process requires a deep grammatical scrutiny of the text depending on the linguistic context [18].

Moves and steps are clearly identifiable parts that organise the ideas in the text. Therefore, the analysis of rhetorical moves is considered as a descriptive method that enables knowing how a text is constructed. Furthermore, this analysis has a clear pedagogical orientation, since it offers apprentices a formal communicative structure to fulfil a social and community function [19]. Subsequently, textual genre analysis is defined as a social activity, aimed at achieving common goals, which involves its members in cultural, professional and/or scientific participation.

This interest in the structural analysis of the text, in university contexts, is corroborated in studies that investigate the moves and steps in various types of academic texts as varied as: practice diaries [20], letters of request for admission to masters' programs [21], case resolution proposals [22], introduction to research articles [23], and research articles [24].

In this line, research emphasises that the training of the university student must include those competences that allow her/him to construct scientific argumentative essays (hereinafter SAE), meaning texts for the purposes of scientific dissemination. This need is based on two premises. In the first place, written argumentation is a basic skill for the communication of knowledge resulting from research. Secondly, written argumentation activates high scientific level mental operations such as search, selection, understanding, reflection, logical reasoning, organisation, synthesis and application to other contexts, and making future forecasts [25,26]. For all these reasons, argumentation is a type of textual genre that is an inalienable part in the construction of scientific articles, as well as a means of initiation in the knowledge of any discipline and in its communicative practices.

However, this described interest contrasts with the results of research in which the absence of a clear structure in the argumentative essays of university students is found [27]. These results are corroborated by the study by Arroyo and Jiménez-Baena [17], in which deficiencies in the formulation of the premise, the reasons against, refutations, citations and bibliographic references are identified. Other studies highlight that, even though there have been improvements after receiving a didactic action, absences of moves that organise the argumentative written discourse continue to be detected [28,29].

For all the above, the need to base the quality of university education to guide the learning of organisation, or structuring, of the scientific argumentative essay is highlighted. To do this, this research aims to investigate what metacognitive and motivational factors are associated with the structure of the SAE, and what this association is like.

## 1.2. Writing Metacognition at University

Writing metacognition refers to the reflection on the writing process in all its complexity, in order to understand it, in addition to the self-regulation and control of such process [30–32], to achieve a communicative objective in a sociocultural, professional and/or scientific context.

From an integrative perspective, the Metasociocognitive Model of Written Composition [33,34] conceptualises writing as a metacognitive interaction of cognitive and sociocultural factors, which are deployed by the action of motivational factors. This interaction is highlighted in the literature on the subject (see for example: [35–39]. Therefore, the importance of checking the level of students' awareness of their writing process globally [40,41] is emphasised, in order to guide learning towards new levels of writing conceptualisation, that give efficiency to written communication.

The techniques used to investigate the writing metacognition of university students and its relationship with other writing dimensions are in a wide range. Thus, Limpo [42] applies the technique of Olive, Kelogg and Piolat [43], which allows the analysis of the temporary organisation and effort used in the planning, transcription and revision of writing process, and verifies how these factors are affected by motivational factors, such as affection and interest.

Karlen [44] develops a questionnaire on metacognitive strategies, finding a correlation with the scores obtained in the production of a text (writing performance). This same technique is used by Csizér and Tankó [45] to inquire into writing self-regulation and its relationship with the motivational factors of writing. Meneses [46] discovers that writing metacognition correlates with self-regulation and writing self-efficacy.

Another applied technique is the online self-report, used by Izquierdo-Magaldi, Renés-Arellano and Gómez-Cash [47] to discover, on the one hand, the metacognitive strategies applied in the development of writing and, on the other hand, the technological resources used by the student in the development of these strategies.

Arroyo [15], applying the interview technique, investigates the metacognition of writing in university students, confirming, on the one hand, the knowledge of planning, transcription and revision of writing and, on the other hand, the low awareness of the control and self-regulation of writing, as well as of sociocultural factors that affect writing. Furthermore, Arroyo and Gutiérrez-Braojos [48] find no significant differences in the writing metacognition of first- and last-year students as they pass through university.

Based on the background, the importance of writing metacognition is highlighted due to its relationship with motivational factors, technological uses, and written products of university students. Additionally, students' metacognitive writing deficiencies are discovered and these do not improve as they pass through the University. Nothing has been studied on how the writing metacognition that the student possesses is associated with the way of structuring the SAE and with the most widespread motivational factor in the literature on the subject—that is, writing self-efficacy.

## 1.3. Writing Self-Efficacy

Writing development at different educational levels demands motivational processes from students, among which are self-efficacy in writing [49]. Self-efficacy is understood as the students' perception of their own ability to achieve specific competencies [50] and the predictive power of self-efficacy on the involvement of students in metacognitive tasks in general has been demonstrated [51]. Furthermore, it has been discovered that self-efficacy in language is associated with self-regulation [52].

More specifically, at university levels, the relationship has been determined between writing self-efficacy and other motivational dimensions such as anxiety [45,53], attitudes towards writing [54,55], apprehension and beliefs [42,56,57], intrinsic, extrinsic motivation and effort [58,59]. In addition, the reviews on regression analysis show that writing self-efficacy is an independent variable for predicting writing achievement [60].

Besides this, university writing self-efficacy is associated with writing metacognition, so Zimmerman and Bandura [61] identify self-efficacy in planning, organising and revising writing,

and, likewise, Andrade, Wang and Akawi [62] demonstrate relationships of writing self-efficacy with revision of the text. Another study demonstrates not only the correlation between self-efficacy and text review, but also the scores obtained in argumentative essay [63].

It can be concluded that writing self-efficacy, at university levels, is a complex motivational construct referring to the feeling of competence to perform a variety of writing tasks, which interrelates with other factors in the writing process. Accordingly, Teng, Sun and Xu [16] discover three dimensions of writing self-efficacy, and their correlation with beliefs towards writing and with the scores obtained in the text. Additionally, Bruning, Dempsey, Kauffman, McKim and Zumbrunn [64] examine three different dimensions in writing self-efficacy, which are validated by Ramos-Villagrasa, Sánchez-Iglesias, Grande-de-Prado, Oliván-Blázquez, Martin-Peña and Cáncer-Lizaga [65]. However, MacArthur, Philippakos and Graham [66], identify a single factor in writing self-efficacy and also show the relationship of writing self-efficacy with the achievement of goals, beliefs and affectivity in writing.

Although there is no complete agreement when it comes to identifying the dimensions of "writing self-efficacy", there is a consensus on its involvement with metacognitive self-regulatory processes, which justifies its inclusion in programs that seek to improve the quality of academic texts [28,58,59,67]. However, the diversity of results in these investigations highlights the need to base didactic interventions according to the real relationships between the different factors that are intended to be promoted.

In short, in search of guidelines that support the quality of education for scientific communication, based on a) the structuring of written argumentation, b) metacognition of writing, and c) argumentative writing self-efficacy, although the literature consulted highlights the importance of these factors in the learning of writing and anticipates associations between them, there are no studies that analyse the relationship between them.

## 2. Methods

To answer the questions raised, this research proposes an exploratory analysis that combines qualitative and quantitative techniques [68].

### 2.1. Participants

The participants in this research are 72 first-year students in a Faculty of Education in southern Spain, specifically at the University of Granada (Andalusia). The age of these students is between 18 and 23 (Average=19.7 SD: 0.98). Of these, 72% are women and 27% men. All voluntarily signed up for this activity as part of the assessment in a subject on their academic curriculum in their native language (Spanish). The enrolled students come from different Andalusian provinces (23% Jaen, 26% Cordoba; 15% Malaga, 36% Granada). This geographic, socio-cultural and economic diversity provides the group with a certain representation.

### 2.2. Instruments

First, a metacognitive questionnaire on the writing process, called Cuestionario Metasociocognitivo (hereinafter CM) is applied. The CM is a written interview consisting of 20 items of the type: Item 18. "When I write, I use a trick or strategy I have discovered to make the text come out right." The items that make up this questionnaire have been designed to extract information about the knowledge, control and self-regulation of the students' cognitive and social factors as writers. The student expresses his/her agreement with each item, 0 being the most negative value and 100 the most positive value. This scale of values was chosen to encourage students to qualify their opinions as much as possible. This range of values between 0 and 100 also makes it possible to compare the responses of this scale with respect to the others used in the research.

The procedure applied for the elaboration of the CM is the discussion group. The procedure followed is the following: (1) the members of the discussion group read and reflect on previous metacognitive writing questionnaires, elaborated by Arroyo and Gutierrez-Braojos, Karlen,

and Meneses [44,46,69]; (2) orally, the most appropriate items are discussed and selected to measure writing metasociocognition; (3) with the conclusions of constructing the CM, reformulating and adapting the wording of some items and adding others.

Second, a scale to measure written argumentative self-efficacy, called the Escala de Autoeficacia (hereinafter EA) is applied. In this case, the scales of MacArthur, Philippakos and Graham [66] were adapted. The EA includes 10 items of the following types: Item 2. "I can write a well-organized argumentative essay", where the student must express his/her agreement with each item, 0 being the most negative value and 100 the most positive value. The items that make up this questionnaire extract information on how the student perceives him/herself when faced with the tasks of writing an argumentative essay.

The validation of both instruments is carried out in a previous study [70] where the internal consistency and reliability of both instruments is verified for a sample of 518 students. The combination of Exploratory and Confirmatory Factor analysis techniques determines the existence of six relevant factors in CM: Factor 1, referring to metacognitive control of writing; Factor 2, referring to the self-regulation of writing; Factor 3, referring to writing planning; Factor 4, referring to writing transcription; Factor 5, referring to writing revision; and, lastly, Factor 6, referring to the writing audience. On the other hand, for AE a single factor is determined. In addition, the stability of both measuring instruments is checked using the test–retest procedure.

Finally, in order to verify the internal consistency of both CM and EA, the Cronbachs' alpha is calculated for the sample of subjects to which this study is applied, obtaining 0.76 for the CM and 0.80 for the EA. According to George and Mallery [71], these values are admissible to accept the reliability of the scales.

Another instrument applied is the Text, which consists of writing a SAE on a given topic, completely independently. All these instruments are available to students online and are applied in a face-to-face session, lasting two hours.

### 2.3. Data Analysis Procedure

First, a qualitative procedure was followed to identify and quantify the rhetorical moves and steps in the students' SAEs. Second, statistical analysis techniques were applied to all the information collected after analysing all the data.

#### 2.3.1. Qualitative Analysis

SAEs are subjected to the content analysis method. For this, a category system is elaborated in a discussion group, based on the proposals of Nussbaum and Kardash, Takao and Kelly and Venables and Summit [72–74] that allows classifying the content of argumentative essays in relation to the rhetorical moves they express. Subsequently, in others studies [17,70,75,76], this system was empirically validated via analysing the argumentative essays of university students.

In the present investigation, an exploration is carried out, coding the SAE of the students in Nvivo11, according to the selected system of categories, by three researchers, applying the argumentation technique to disagreements. After this exploration, the steps that describe the rhetorical moves of the SAE are reformulated and others are added, leaving the category system as it appears in Table 1. Subsequently, a second exploration of all the SAEs is carried out and the counting rules are applied. That is, the number of times each rhetorical step appears in all SAEs. The frequency of each move is obtained by adding the frequencies of all the steps contained in that move.

**Table 1.** Structure of scientific argumentative essay (SAE).

| Rhetorical Moves | Description of Each Move in Steps |
| --- | --- |
| Introduction | Includes the following steps: presentation of the topic, relevance of the topic, innovations and citations |
| Premise | Includes the following steps: formulation of the claim, definition of premise concepts and citations |
| Argument | Includes the following steps: reasons in favour of the premise, reasons against, rebuttals, definition of new concepts, expert quotes and research quotes |
| Conclusion | Includes the following steps: synthesis of reasons, definitive reason, application, projection, appointments |
| Bibliography | Bibliographic References |

Source: Adapted from Arroyo and Jiménez-Baena [17] (p. 358).

### 2.3.2. Quantitative Analysis

The result of the coding and counting procedure, described in the previous section, allows defining six structural variables of scientific argumentative writing: Introduction, Premise, Argument, Conclusion and Bibliography (described in Table 1). In addition, a sixth variable is defined: Rhetorical Moves, which is obtained with the total sum of the frequencies obtained in each move. With the partial and total frequencies, the arithmetic mean is calculated. Aside from this, the responses of the students in CM and AE, gave rise to the variable: Writing Metacognition and Argumentative Self-efficacy, respectively. The value of these variables consists of the arithmetic mean of the scores of all the items. In the three variables, the arithmetic means obtained are divided by 10; therefore, all of them take values in the interval 0–10.

For statistical calculations, the R Commander software is applied in two stages. In the first, the Pearson correlations are calculated, in the second, the relationships that are significant after the analysis of the correlations are explored. For this, linear regressions are applied.

Since there is no previous research to establish an a priori hierarchy among the variables considered, the optimal linear model that explains this hierarchy is sought. A step-by-step regression procedure is followed. It starts from the model that predicts the selected dependent variable, starting with the one with the highest correlation and, progressively, predictive variables are added to increase the explanatory power of the models obtained. This explanatory power is measured using the corrected linear determination coefficients $R^2$ and the information functions of Akaike and Schwarz [77,78]. To do so, the model with the highest corrected $R^2$ value (the one that explains the most variation in relation to the number of variables it contains) and the one that provides the least information function (the one that contains the least unexplained variation) are selected.

Finally, the dependence of the observed results with respect to factors not directly measured is contrasted. For this, a confirmatory analysis based on structural equations is proposed [79], which is determined by the data obtained and the relationships previously found. This analysis is performed using the RCommander lavaan package.

## 3. Results

The descriptive analysis of the main variables of this research shows that, on the one hand, Argumentative Self-efficacy yields an average score of 6.93 with a minimum of 5.4 and a low dispersion of the data (Pearsons' variation coefficient of 0.14 and range of 4.6 out of 10), highlighting the high levels of this variable.

The results of Argumentative Self-efficacy are similar to those found in Writing Metacognition, where average values around 7.5 and high representativeness of the position measurements are found (Pearsons' variation coefficient of 0.12 and range of 4.2 over 10).

On the other hand, the count of Rhetorical Moves shows more heterogeneous results. Firstly, there is a low frequency of Premise (a maximum value of 4, with an average of 78 per student) or Bibliography (maximum value of 1). However, higher frequencies are seen in Introduction (an average of 4.85) and, more widely, in Argument and Conclusion (with average values of 2.78 and 2.17, respectively, and coefficients of variation around 1, compared to. 62 of Introduction).

## 3.1. Analysis of Correlations Between Variables

Table 2 presents the Pearson correlations between the different variables. It is observed that Writing Metacognition and Rhetorical Moves are more strongly correlated with each other (0.397) than with Argumentative Self-efficacy (0.282 and 0.212, respectively).

**Table 2.** Pearson correlations between the study variables.

| Variable | 1 | 2 | 3 | 4 | 5 | 6 | 7 |
|---|---|---|---|---|---|---|---|
| 1. Self-efficacy | | | | | | | |
| 2. Metacognition | 0.282 * | | | | | | |
| 3. Rhetorical Moves | 0.212 | 0.397 ** | | | | | |
| 4. Introduction | 0.049 | 0.070 | 0.664 ** | | | | |
| 5. Premise | 0.237 * | 0.339 ** | 0.546 ** | 0.063 | | | |
| 6. Argument | 0.195 | 0.331 ** | 0.749 ** | 0.168 * | 0.457 ** | | |
| 7. Conclusion | 0.184 | 0.437 ** | 0.756 ** | 0.270 * | 0.480 ** | 0.463 ** | |
| 8. Bibliography | 0.163 | 0.384 ** | 0.467 ** | 0.324 ** | 0.406 ** | 0.276 * | 0.233 * |

Note: * = indicates that the *p*-value is less than 0.05; ** = indicates that the *p*-value is less than 0.01.

This trend is corroborated by observing the relationships between the different rhetorical moves. In particular, the correlations of Argumentative Self-efficacy have values between 0.049 (with Introduction) and 0.237 (with Premise), while Writing Metacognition keeps correlations between 0.07 (with Introduction) and 0.437 (with Conclusion). By attending to the relationship between the different rhetorical moves; the lowest correlations are found in Introduction with Premise (0.063) and Argumentation (0.168). In turn, the highest correlations correspond to Conclusion, with Premise (0.480) and with Argumentation (0.463).

Each of the different rhetorical moves show high correlations (between 0.467 and 0.756), with the variable Rhetorical Moves, as would be expected, and the highly significant correlations (p-value is less than 0.01) with Writing Metacognition standing out. On the other hand, only Premise correlates significantly with Argumentative Self-efficacy (p-value is less than 0.05).

## 3.2. Analysis of Linear Regression Models

The results of the linear regression contrasts based on Argumentative Self-efficacy can be seen in Table 3. The regression of the Writing Metacognition variable shows a significant relationship (*p*-value = 0.0016) that allows explaining 6.62% of the variation of this variable from Argumentative Self-efficacy. On the contrary, the model for the variable Rhetorical Moves was not significant at level 0.05 (*p*-value = 0.0735), which denotes the lower explanatory power of Argumentative Self-efficacy over Rhetorical Moves—that is, the way in which the student organises the argumentative essay.

**Table 3.** Linear regression models based on Argumentative Self-efficacy as a predictor variable.

| Variable Explained | (a[1]) | (b[2]) | $R^2$ | *p*-Value |
|---|---|---|---|---|
| Metacognition | 5.65 | 0.26 | 0.0662 | 0.0016 |
| Rhetorical Moves | 1.15 | 10.39 | 0.0314 | 0.0735 |

Note: [1] Denotes the independent term; [2] The coefficient of the variable Self-efficacy of the model. The value of $R^2$ multiplied by 100 expresses the percentage of the variation explained by the model.

Therefore, based on the results, the analysis focuses on the explanatory models for the rhetorical moves variables based on the Writing Metacognition variable (Figure 1). The regression of Writing Metacognition explains 14.59% of the variation of Rhetorical Moves. The p-values in Table 4 show that the linear regression is significant for every rhetorical move, except for the Introduction (*p*-value of 0.56). Specifically, the models with the greatest explanatory power are Conclusion ($R^2$ = 0.1797) and Bibliography ($R^2$ = 0.1354).

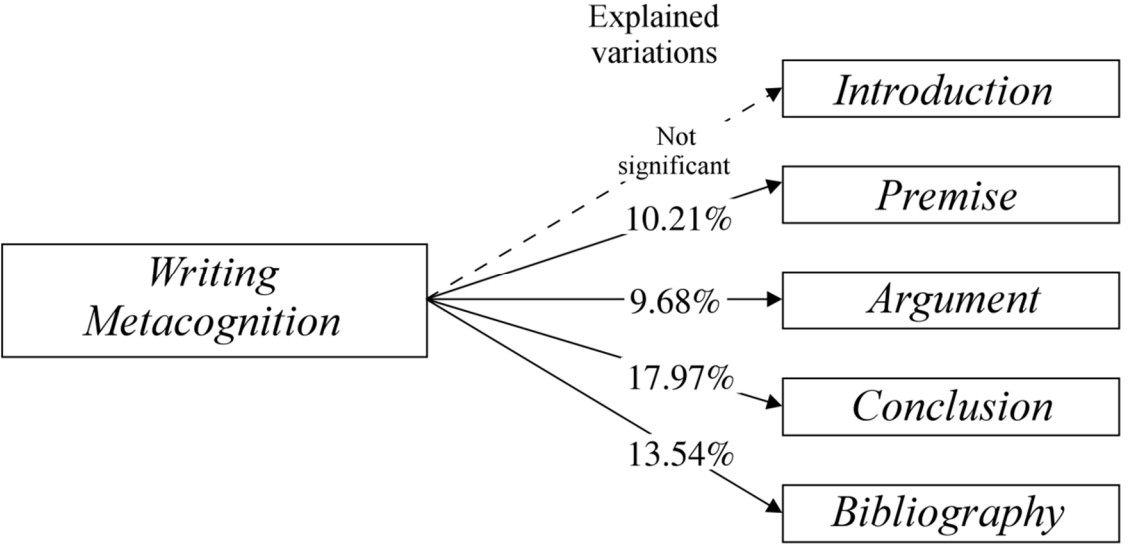

**Figure 1.** Regression model based on Writing Metacognition.

**Table 4.** Regression models of the different structural movements based on Writing Metacognition as a predictor variable.

| Variable Explained | a | b | $R^2$ | *p*-Value |
|---|---|---|---|---|
| Rhetorical Moves | 9.88 | 0.15 | 0.1459 | 0.0005 |
| Introduction | 3.16 | 0.23 | 0.0019 * | 0.56 |
| Premise | −1.39 | 0.29 | 0.1021 | 0.0036 |
| Argument | −4.67 | 1.01 | 0.0968 | 0.0045 |
| Conclusion | −5.92 | 1.10 | 0.1797 | 0.0001 |
| Bibliography | −1.06 | 0.17 | 0.1354 | 0.0008 |

Note: * = indicates that the *p*-value is less than 0.05.

Based on these results, Introduction is eliminated, since the correlation with Writing Metacognition is not significant, and, on the other hand, Conclusion is selected as the first predictor variable. Subsequently, the variables Bibliography, Premise and Argument are added to the model, in the order mentioned. The results of this procedure can be seen in Table 5.

**Table 5.** Step-by-step regression and optimal model.

| Predictor Variable | a | b | $R^2$ | AIC | BIC |
|:---:|:---:|:---:|:---:|:---:|:---:|
| Model 1 **: | 0.17 | | 0.1797 | 181.04 | 187.87 |
| Conclusion | | 7.00 | | | |
| Model 2 **: | 6.92 | | 0.2544 | 175.13 | 184.43 |
| Conclusion | | 0.15 | | | |
| Bibliography | | 0.67 | | | |
| Model 3 **: | 0.90 | | 0.2461 | 176.88 | 188.26 |
| Conclusion | | 0.14 | | | |
| Bibliography | | 0.62 | | | |
| Premise | | 0.07 | | | |
| Model 4 **: | 6.86 | | 0.2416 | 178.24 | 191.90 |
| Conclusion | | 0.12 | | | |
| Bibliography | | 0.60 | | | |
| Premise | | 0.03 | | | |
| Argument | | 0.04 | | | |

Note: ** = indicates that the *p*-value is less than 0.01.

The model based on Conclusion and Bibliography as explanatory variables is significant at 99.9% confidence (*p*-value of 0.00001489) and provides an explained variation of 25.44%, which is the highest found during the step-by-step regression procedure (Figure 2 and Table 5). Likewise, information functions were obtained from Alkaike and Schwartz of 175.33 and 184.43 respectively, the lowest of all the models considered. Similarly, the inclusion of the Premise and Argument variables does not provide explanatory power to the model, since adding them to the model causes the corrected $R^2$ values to decrease and both information functions to increase.

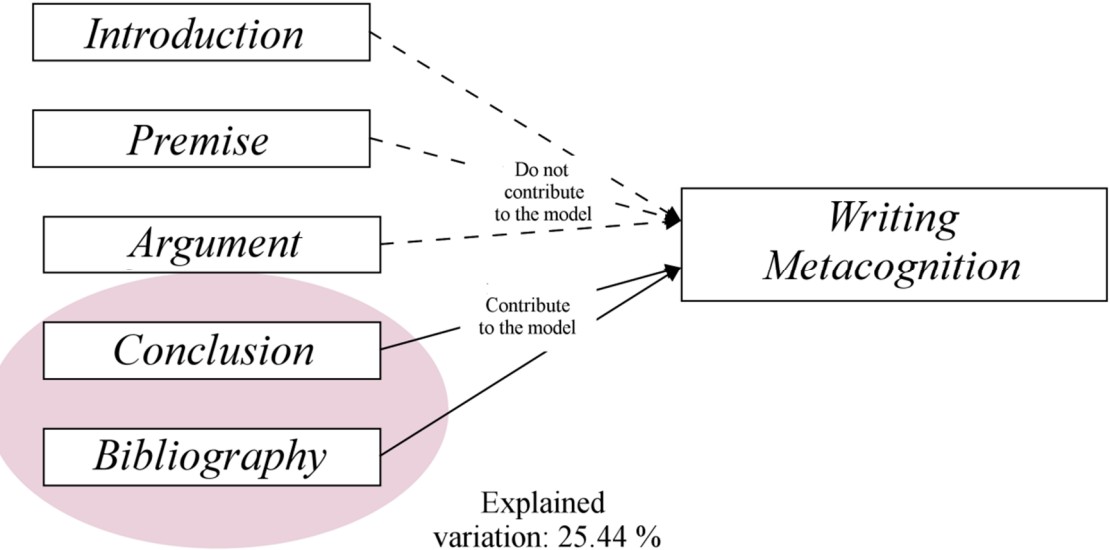

**Figure 2.** Optimal regression model.

*3.3. Analysis of Structural Equation Models*

The models obtained that explain greater variations are those that relate the variables Writing Metacognition with the rhetorical moves Conclusion and Bibliography. This fact points to the existence

of a motivational factor different from Argumentative Self-efficacy, which explains the connection between the metacognitive writing knowledge of university students and the inclusion of these rhetorical moves when they organise their argumentative essays. To corroborate this hypothesis, a confirmatory analysis is proposed. Table 6 shows the estimated parameters of the model.

**Table 6.** Estimated parameters of the model that explains the relevant variables in the regression models.

| Variable | Estimation Parameter | Estimation Error |
|:---:|:---:|:---:|
| Metacognition | 1 | 0.23 |
| Conclusion | 1.521 | 3.84 |
| Bibliography | 0.239 | 0.133 |

The weighted least squares estimation method gives rise to an adjustment model with $R^2$ = 16,929 and 3 degrees of freedom (*p* value equal to 0.001). These data provide a ratio $R^2/gl$ = 5.643, so the model is at the limit of the acceptable [80]. On the other hand, the incremental measures of adjustment support the acceptance of the model, since values of CFI = 1 and TLI = 1 have been found. Furthermore, the mean approximation errors by degrees of freedom found yield values RMSEA = SRMR = 0. In conclusion, the model obtained to explain the variables of Writing Metacognition, Conclusion and Bibliography can be accepted from a motivational factor.

## 4. Discussion

The descriptive analysis indicates that the students express with variable frequencies all the rhetorical moves of the SAE, which denotes certain ability in the task. However, students include few Bibliographic References and the presence of other rhetorical moves is variable. This means that a common model cannot be identified when structuring the SAE, although the inclusion of the Introduction move is highlighted in the entire argumentative essay. These results coincide with other investigations where there is a lack of concordance in the rhetorical moves that are identified in the analysis of textual structures [23,24].

In relation to the question: is there a relationship between the way of structuring the argumentative essay, writing metacognition and the writers´ argumentative self-efficacy by university students? First, a relationship between argumentative self-efficacy and writing metacognition is discovered. These results coincide with the findings of other studies [52,64] where the relationship between writing self-efficacy and self-regulation of the writing process (which is a metacognitive dimension) are highlighted. However, in this case, the participants are high school students. Likewise, the results of this research coincide with those of another [81], where the correlation between the knowledge of the writing process (which is also a metacognitive dimension) and the writing self-efficacy in university students is verified. In addition, it is demonstrated that this correlation increases after the application of a program to teach writing of argumentative essays at University.

Furthermore, in this research, writing metacognition presents higher correlations with the rhetorical moves of the SAEs than with the writing argumentative self-efficacy. This finding is highlighting the metacognitive perspective over the motivational perspective, in relation to the argumentative writing performance. These results are supported by the study by Teng, Sun and Xu [16] where moderate correlations between the dimensions of writing self-efficacy and the scores given by evaluators to argumentative essays are shown. For their part, MacArthur, Philippakos and Graham [66] discover that self-efficacy does not significantly correlate with the scores given to argumentative essays. Likewise, in the research by Sanders-Reio, Alexander, Reio and Newman [56] it is found that the correlation between self-efficacy and essay scores is lower than that presented in other studies with students of lower academic levels [82,83]. However, the results described contrast with the findings in other investigations [58,84], where writing self-efficacy stands out as a factor associated with the scores given to the essay of university students.

It is important to point out that in none of the mentioned studies the rhetorical moves of the text produced by the students are analysed as a measure of their writing performance. Therefore, the aforementioned results may be conditioned, either, by the different measures, or by the different academic contexts in which the research takes place.

In any case, in the present investigation it can be concluded that the argumentative self-efficacy of writing is not the most optimal motivational dimension to predict the way in which university students structure their argumentative essays. In other words, having greater confidence in your own writing skills does not guarantee greater ability to structure your essay. This finding, in some way, contradicts one of the starting premises of the literature, which considers self-efficacy as the motivating axis of any writing process [49]. However, this contradiction may be indicating "that motivation-related variables may not play the same key role in writing performance in expert writers as it has in novice writers" [82] (p. 120).

On the other hand, the results of the present investigation discover insignificant correlations of the rhetorical moves with argumentative self-efficacy, except with Premise. This means that students who feel competent in scientific written argumentation are able to formulate a premise. It also indicates that the inclusion of the rest of the rhetorical moves of the SAE may be motivated by some other unmeasured motivational factor.

Regarding the second question: is it possible to identify any of the measured variables as factors that explain the others and/or anticipate new factors? The regression models based on Writing Metacognition significantly explain the following rhetorical moves: Premise, Argument, Conclusion and Bibliography (See Figure 1 above), which are argumentative genre-specific rhetorical moves to achieve purposes within a scientific context [10].

By contrast, writing metacognition presents no explanatory power about rhetorical moves inclusion: Introduction; data that contrasts with the high frequency of this move in the students' SAE. This fact indicates, in the first place, that university students recognise the importance of introducing the topic to be developed in an essay, coinciding with Brown and Marshall [10]. Secondly, the importance given to the Introduction can be explained because this move is defined by steps of the explanatory genre, which is a more common type of text in academic writing. Therefore, the demand for reflection that writing metacognition requires does not explain the inclusion of this rhetorical move, but it does explain the inclusion of the specific rhetorical moves mentioned in the previous paragraph. By virtue of these results, it can be affirmed that metacognitive knowledge about writing favours the inclusion of certain rhetorical movements in the argumentative essay related to the body of the text, the conclusion and the bibliography, but does not contribute to the contextualisation of the subject matter.

On the other hand, the optimal regression model (see Figure 2 above), highlights that it is the inclusion of rhetorical moves Conclusion and Bibliography which predict the students´ writing metacognition. This means that the practical ability to build these moves is what predicts the knowledge, control and self-regulation of the writing process.

These findings are explained because including the Conclusion and Bibliographic References moves, when organizing an SAE, implies expressing the following steps: synthesis of reasons, definitive reason, application and projection, and quotations. All these steps require search, inquiry, selection, reflection and synthesis skills on ideas and their expression, and it is presumable that these skills are also applied to the knowledge, control and self-regulation of the writing process itself—that is, to writing metacognition.

Regarding the third question: What are the implications for a quality education at the University that are inferred from the relationships found? Firstly, it is verified that the students do not have a common organisational model when it comes to building SAE, beyond the scarcity of Bibliographic References and the greater presence of the move Introduction. These data evidence the need for a higher educational quality in the University that guides the construction of SAE. This educational quality should promote students' likely to (a) formulate a clear premise based on well-defined concepts,

(b) build arguments based on the existing literature on the subject to be treated, and (c) to include proper references.

The results also indicate that writing metacognition is associated with the way the student structures the SAE. This finding supports the didactic criterion of promoting awareness or reflection of the writing process itself in university teaching to achieve high-quality quotas [85,86].

Furthermore, the results confirm that it is the practical ability to construct a Conclusion, and include Bibliography, which predicts the writing metacognition. This finding has a crucial pedagogical implication, since it is highlighting the need to design programs to enhance the SAE, focused on structuring them and, above all, on the inclusion of those moves that require more investigation and synthesis, as has been noted above.

Finally, the minor relationship that written argumentative self-efficacy shows with the expression of rhetorical moves of the SAE suggests the need to investigate other factors that explain the relationship between writing metacognition and rhetorical moves Conclusion and Bibliography, as evidenced in the structural equation model (Figure 3). The acceptance of this model suggests that the relationships found in the regressions may have their origin in factors related to the intrinsic motivation of the students.

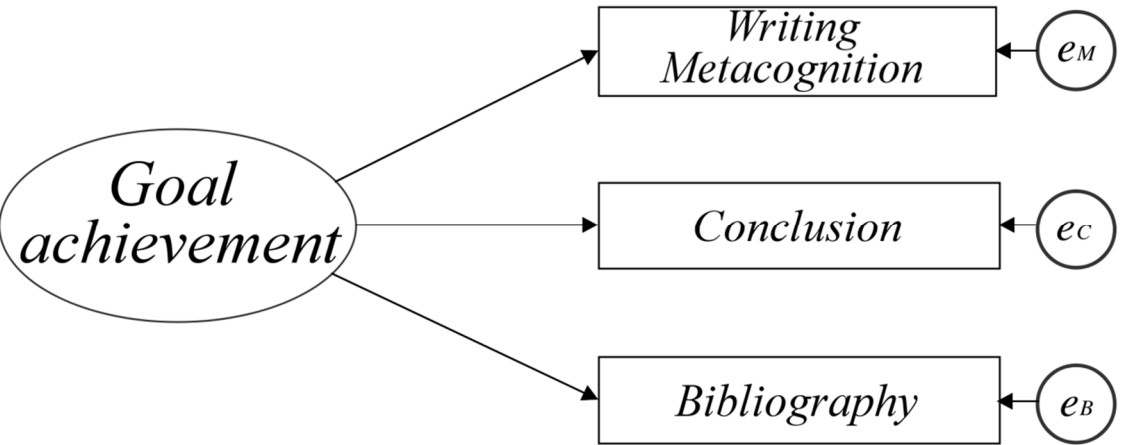

**Figure 3.** Motivational factor that can explain the variables of relevance in the regression models.

The assumption that this factor can be motivational is based on the conceptualisation of writing as a process of an affective nature, as well as cognitive and structuring of meaning [87]. In this line, the literature points to the need to promote the functionality or authenticity of written communication for its application to professional life [88]—that is, the orientation of writing towards the achievement of objectives. Thus, goal achievement is suggested as a more influential motivational factor than self-efficacy in argumentative writing at university levels [41]. Therefore, a new research challenge arises from the present study, which is to demonstrate this hypothesis.

## 5. Conclusions

In conclusion, in the field of the quality of higher education, as a sustainable development objective, written communication must be guided to meet the criteria of scientific dissemination [89]. Furthermore, writing at this level needs to anticipate efficient written communication with a specific audience, creating the opportunity to share sustainability goals in a specific field of knowledge.

More specifically, effective written argumentative communication is an undeniable challenge for the quality of higher education by 2030. Achieving this effectiveness, in turn, requires applying didactic designs, based on basic metacognitive, motivational, and grammatical skills, that allow guiding the argumentative learning writer for scientific dissemination, with supports and resources, appropriately provided [90].

Consistent with this line, this research promotes the quality of university education for learning scientific written argumentation. The need to train in this basic university skill is giving rise to

the proliferation of works on the development of specific programs [91–93], some of them using technological systems [94,95]. However, the variable results highlight the need to base their educational quality on factors that explain the communicative effectiveness of the texts that are intended to be promoted.

Regarding the factors that can determine this effectiveness, the literature review highlights writing motivation, writing metacognition, and adequate text structuring. The main novelty of this research is that the relationships between these factors are analysed, taking into consideration the expression of the different rhetorical moves of the SAE.

Thus, it is concluded that it is the practical ability to structure a Conclusion and Bibliography when writing an SAE which predicts the students´ writing metacognition. On the other hand, the minor relationship that argumentative self-efficacy shows with the expression of rhetorical moves, compared to the writing metacognition, highlight the need to consider another motivational dimension that is driving the learning of argumentative essay at the university level, a hypothesis that is confirmed with the structural equation model.

All these findings make it possible to establish a series of criteria for the quality of university education, these are as follows: (i) to enhance the orientation to professional–scientific goals as a motivational dimension for the construction of argumentative essays, evaluating their effects; (ii) to promote the metacognitive dimension in the construction of argumentative essays; (iii) to promote learning experiences for the construction of argumentative essays on specific topics, where the structuring of ideas into moves and steps is guided with appropriate supports and resources; to influence those moves that require inquiry, reflection and synthesis, such as Conclusion and Bibliography.

However, these conclusions are subject to the limitations of the present investigation. Firstly, the difficulty of achieving a representative sample in current educational contexts must be pointed out. Furthermore, research participants present homogeneous socio-economic circumstances and academic traits. These two factors impair the external validity of the results obtained, although the procedures and conclusions may be transferable in any context.

Consequently, this research highlights the need to further explore the explanatory power of the factors analysed in wider contexts, in order to support the educational quality of written communication in higher and medium education worldwide and to ensure the inclusion of people with linguistic diversity.

To this end, the projects that are being developed contemplate didactic experiences in different languages with university students. The aim is to implement high-quality education, which follows the criteria stated above, to develop students' written argumentation that supports the dissemination of sustainable knowledge. Finally, another line of research which is currently developing is the analysis of factors affecting writing in other genres, for their teaching in different educational levels, as well as for the inclusion of people with functional diversity and immigrants.

**Author Contributions:** R.A.G., J.M.G. and J.d.l.H.-R. conceived the hypothesis of this study. R.A.G. and J.d.l.H.-R. participated in data collection. J.M.G. and J.d.l.H.-R. analysed the data. All authors contributed to data interpretation of statistical analysis. R.A.G., J.M.G. and J.d.l.H.-R. wrote the paper. All authors have read and agreed to the published version of the manuscript.

**Funding:** This research has been carried out with funding from the Autonomous Community of Andalusia (Research Group ED.INVEST, HUM356), the University of Granada (Quality Innovation and Prospective Unit, project 373) and the Ministry of Economy, Industry and Competitiveness (DER2017–89623-R project).

**Conflicts of Interest:** The authors declare no conflict of interest.

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
