# Peer review of "The 2030 Challenge in the Quality of Higher Education: Metacognitive, Motivational and Structural Factors, Predictive of Written Argumentation, for the Dissemination of Sustainable Knowledge"

_sustainability, doi:10.3390/su12198266_

Round 1

Reviewer 1 Report

This paper deals with effective writing competence at higher education level and, investigates the factors that predict effective written communication.

It seeks to determine the effectiveness of written argumentative communication by establishing correlations between rhetorical moves (the way written argumentation is organized, metacognitive variables regarding the writing process and c) motivation.

The introduction does not present the cognitive construct well to readers who are not from the field. I missed basic definition of motivation: "Motivation is the process whereby goal-directed activity is initiated and sustained”( Motivation in Education. Schunk, Pintrich & Meece, 2008)

For Metacognition and Meta cognitive processes -  see Flavell, 1976; Brown, 1987

For Argumentation see Tulmin, as a classical perspective, and Erduran and Osborne, 2004, 2006

It is also interesting to refer to logic fallacies in students' writing

Participants, methods and analysis are clearly described

With regard to the identified rhetorical moves in students' written texts:

I strongly disagree with the description of the move 'argument' – the authors describe it as: ' "reasons in favor of the premise, reasons against, rebuttals, definition of new concepts, expert quotes and research quotes"

To my opinion this identification lacks the basic part of an argument: a claim, and evidence that support it. These are fundamental to writing and identifying them may change the results of the correlation analysis.

Table 2. Pearson correlations between the study variables: I suggest including in the upper headline the name of the move and not only the number.

Results are clearly described and explained, so are the research limitations and application.

The authors suggest an optimal model of correlation between rhetorical moves and writing meta cognition and a model to explain how the motivational is correlated with variables of relevance in the regression models.

The models are supported by the statistical analysis and are based on correlation but it lacks the detailed causal explanation behind the regression.

Author Response

Attached file with revisions

Thank you.

Best wishes.

Reviewer 2 Report

As for research area, authors discuss the quality of higher education, as a sustainable development objective where written communication must be guided to meet the criteria of scientific dissemination.

Their research promotes the quality of university education for learning scientific written argumentation. They highlight the importance of appropriate didactic approach which would support basic metacognitive, motivational, and grammatical skills, that allow guiding the argumentative learning writer for scientific dissemination.

Authors provide readers with an insight into the topic via thoughtfully selected sources.

Description of the research and its aims are clearly formulated via research questions.

Applied methods: Content analysis, linear regression models, exploratory and confirmatory analysis.

Applied instruments: a metacognitive questionnaire on the writing process, called Cuestionario Metasociocognitivo (hereinafter CM) and a scale to measure written argumentative self-efficacy, called the Escala de Autoeficacias.  Verification of the internal consistency of metacognitive questionnaire  and a scale to measure written argumentative self-efficacy was made via calculation of the Cronbach's alpha. For statistical calculations, the R Commander software was applied to calculate Pearson correlations and linear regressions.

Authors proceed in a systematic way, explain individual terms, steps and applied methods which enables readers smoothly go through the paper.

The only place which I find little bit confusing is on the page 5 – where authors describe an applied instrument the Cuestionario Metasociocognitivo that is a questionnaire filled in a written form and then authors mention a discussion group “The procedure applied for the elaboration of the CM is the discussion group, based on previous cognitive interviews, elaborated by Arroyo and Gutierrez-Braojos, Karlen, and Meneses [36,38,63]”. Do respondents also talk or is the research run purely on written basis? Authors refer to sources which are not for the reader at hand with the first reading so an explanatory sentence might be fine.  

Author Response

(The authors gave the same response as above.)

Reviewer 3 Report

The article entitled "The 2030 challenge in the quality of higher education: metacognitive, motivational and structural factors, predictive of written argumentation, for the dissemination of sustainable knowledge" presents a significant research problem. The problem is well approached, the theoretical approach is correct, and both the discussion and the conclusions are appropriate and adjusted to the research problem. However, the work presents a critical problem related to the representativeness of the sample. To solve this problem, the authors should justify the sample collection and that the research problem can be addressed with the collected sample. Some suggestions are raised.

Title.

Perhaps the title should be rethought to make it more specific and understandable.

  1. Introduction.

"In this context ..." it might be useful to move this paragraph into the conclusions.

"Therefore, this research aims ..." it might be useful to indicate the specific research objective outlined below in addition to the research questions.

1.2. Writing metacognition at university

"... verifies how these factors are affected by motivational factors" what type of motivational factors are referred?

  1. Method

2.1. Participants

As has been pointed out, the biggest problem in the research is the representativeness of the sample. Besides, it would be convenient for the sample to be described in more detail, pointing out other types of variables that identify students' sample.

2.2. Instruments

It might be useful to present the questionnaire as a whole, for example, in an appendix.

"0 being the most negative value and 100 the most positive value", check if the value 100 is appropriate and, in any case, explain and justify the type of scale.

" Escala de Autoeficacias," check the name of the scale and justify it through bibliographic references.

2.3. Data analysis procedure

"... statistical analysis techniques were ..." it would be convenient to indicate the statistical techniques used and their justification.

  1. Discussion

The set of figures presented in this section should be added in the results section, as they correspond to the research model's estimation. In this section, the results of the model should be discussed.

Author Response

(The authors gave the same response as above.)
